# Random terpolymer based on thiophene-thiazolothiazole unit enabling efficient non-fullerene organic solar cells

Jingnan Wu[1], Guangwei Li[1], Jin Fang[1], Xia Guo[1], Lei Zhu [2], Bing Guo[1], Yulong Wang[1], Guangye Zhang [3], Lingeswaran Arunagiri [4], Feng Liu [2], He Yan[4✉], Maojie Zhang [1✉] & Yongfang Li[1,5]

Developing a high-performance donor polymer is critical for achieving efficient non-fullerene organic solar cells (OSCs). Currently, most high-efficiency OSCs are based on a donor polymer named PM6, unfortunately, whose performance is highly sensitive to its molecular weight and thus has significant batch-to-batch variations. Here we report a donor polymer (named PM1) based on a random ternary polymerization strategy that enables highly efficient non-fullerene OSCs with efficiencies reaching 17.6%. Importantly, the PM1 polymer exhibits excellent batch-to-batch reproducibility. By including 20% of a weak electron-withdrawing thiophene-thiazolothiazole (TTz) into the PM6 polymer backbone, the resulting polymer (PM1) can maintain the positive effects (such as downshifted energy level and reduced miscibility) while minimize the negative ones (including reduced temperature-dependent aggregation property). With higher performance and greater synthesis reproducibility, the PM1 polymer has the promise to become the work-horse material for the non-fullerene OSC community.

[1] Laboratory of Advanced Optoelectronic Materials, College of Chemistry, Chemical Engineering and Materials Science, Soochow University, 215123 Suzhou, China. [2] Department of Physics and Astronomy and Collaborative Innovation Center of IFSA (CICIFSA), Shanghai Jiao Tong University, 200240 Shanghai, China. [3] eFlexPV Limited, Flat/RM B, 12/F, Hang Seng Causeway Bay BLDG, 28 Yee Wo Street, Causeway Bay, Hong Kong, China. [4] Department of Chemistry, Hong Kong Branch of Chinese National Engineering Research Center for Tissue Restoration & Reconstruction, Hong Kong University of Science and Technology (HKUST), Clear Water Bay, Kowloon, Hong Kong, China. [5] Beijing National Laboratory for Molecular Sciences, CAS Key Laboratory of Organic Solids, Institute of Chemistry, Chinese Academy of Sciences, 100190 Beijing, China. ✉email: hyan@ust.hk; mjzhang@suda.edu.cn

Non-fullerene organic solar cells (OSCs) have been attracting increased research attentions[1–4], with the development of high-performance non-fullerene acceptors including Y6, ITIC, and their derivatives[5–14]. Meanwhile, the choice of a suitable donor polymer is also crucial for achieving efficient OSC devices[15–20]. One of most commonly used donor polymer is Poly[(2,6-(4,8-bis(5-(2-ethylhexyl-3-fluoro)thiophen-2-yl)-benzo [1,2b:4,5-b′]dithiophene))-alt-(5,5-(1′,3′-di-2-thienyl-5′,7′-bis(2-ethylhexyl)benzo[1′,2′-c:4′,5′c′]dithiophene-4,8-dione)][21], short-named as PM6, which can generate highly efficient devices when combined with state-of-the-art acceptors[22–25]. Despite of its dominance in the academic community, its performance is highly sensitive to the molecular weight of the material. This causes serious batch-to-batch reproducibility issue and also limits the application of high-performance OSCs to industry. Therefore, it is crucial to develop a high-performance donor polymer with high performance and greater synthetic reproducibility.

To develop alternative donor polymers, one strategy is to synthesize random terpolymers by incorporating a third component (such as an acceptor unit) to form a Donor-Acceptor1-Donor-Acceptor2 (D-A$_1$-D-A$_2$)-type polymer[26–30]. For instance, an ester-substituted thiophene unit has been incorporated into PM6 backbone to downshift the molecular energy level and broaden the absorption, leading to simultaneously enhanced short-circuit current density ($J_{sc}$) and open-circuit voltage ($V_{oc}$) in the device[31]. To address poor phase separation and optimize the morphology, s-tetrazine-based terpolymers have also been developed to suppress the aggregation tendency of the polymer, thus improving the efficiency up to 16.35% in the resulting devices[32]. While the terpolymer strategy has shown some positive effects, it is generally believed that the irregular polymer backbone can cause negative effects on molecular stacking and charge transport of the corresponding polymers[33–36]. Therefore, it is important to construct a terpolymer such that the negative effects of the random nature are minimized while the beneficial effects of the terpolymer strategy can be realized in OSC devices.

In this article, we report a high-performance terpolymer (named PM1) that enables high OSC efficiencies (near 17.6%) and, more importantly, great device reproducibility and batch-to-batch synthetic reproducibility. Our polymer can be synthesized via a random ternary polymerization strategy by incorporating 20% of thiophene-thiazolothiazole (TTz) building block into the state-of-the-art PM6 polymer backbone to obtain D-A$_1$-D-A$_2$-type terpolymers. We carry out morphological and electronic characterizations on the polymer and show that the TTz unit can introduce several effects (either positive or negative): (1) the TTz unit exhibits a near-perfect co-planar structure which could enhance the crystallinity of polymer and charge mobility; (2) the addition of the TTz unit can adjust the miscibility between the polymer and non-fullerene acceptor to achieve more favorable phase interface for high fill factor (FF); (3) the TTz unit also has some negative effects on the temperature-dependent aggregation (TDA) property of the polymer, which is important to achieve optimal morphology and great OSC performance. Overall, it was found that the terpolymer with 20% of the TTz unit is the best one, as it can maintain the positive factors of TTz unit (better crystallinity and charge mobility and higher domain purity), while minimizing the negative impact of TTz unit. With high performance and great reproducibility, we expect the PM1 polymer to be widely used in the OSC community in the near future.

## Results

**Materials design and properties**. As shown in Supplementary Fig. 1, the target polymers of PM6, PM1, PM2, and PBFTz were prepared via the Stille coupling reaction, and detailed synthetic procedures and characterizations can be found in the Supplementary Information. All these four polymers have good solubility in common organic solvents like chloroform and chlorobenzene at room temperature. Basic properties of the polymers, such as the values of number-average molecular weight ($M_n$), are shown in Supplementary Table 1. All the polymers exhibit good thermal stability with decomposition temperatures ($T_d$) at 5% weight loss above 400 °C revealed by thermogravimetric analysis (Supplementary Fig. 2). And the corresponding differential scanning calorimetry (DSC) measurement results are shown in Supplementary Fig. 3, but there are no clear endothermic peak and exothermic peak in the DSC thermogram. In addition, the energy levels of these polymers were estimated from cyclic voltammetry (CV)[37] (Supplementary Fig. 4). As shown in Fig. 1b, as along with the TTz ratio increasing, the energy levels of terpolymers show relatively downshifted compared to that of PM6 and the results were also confirmed by density functional theory calculations (DFT)[38] (Supplementary Fig. 5). Owing to the stronger electronegativity of the sp$^2$-nitrogen atom on TTz unit, the highest occupied molecular orbital (HOMO) levels of the terpolymers efficiently decreased[39–41], which could be beneficial to achieve higher $V_{oc}$ and lower energy loss ($E_{loss}$) in OSCs.

Figure 1c and Supplementary Fig. 6a present the UV-vis absorption spectra of the four polymers in chloroform solutions and solid-state films, respectively. It is observed that there is no obvious difference in the absorption edge from the solution to the film state, implying the strong pre-aggregation of these polymers in the solution state. Besides, with the increasing ratio of TTz, the absorption edges are gradually blue-shifted and 0–0 transition peak is weakened in both neat and blend films, due to the reduced intermolecular stacking push-pull effect and the destruction of aggregation by the random terpolymerization strategy, respectively. Meanwhile, the introduction of a certain amount of TTz content also leads to enhanced absorption coefficient in blend film with the acceptor Y6 (Supplementary Fig. 6c), which should have positive effects on improving the light harvesting ability of the OSCs.

**OSC device performance**. The photovoltaic properties of four donor polymers were studied by fabricating OSCs with a conventional structure of glass/ITO/PEDOT:PSS/polymer:Y6/PFN-Br/Ag. OSCs were optimized by tuning the donor:acceptor (D:A) ratio and changing the content of the solvent additive (chloronaphthalene, CN) to get the best photovoltaic performance (Supplementary Figs. 7 and 8 and Supplementary Tables 2–4). All active layers were formed by spin-coating at room temperature with a photoactive solution at 25 °C on unheated substrates. The current density–voltage ($J$–$V$) and the photovoltaic parameters of the devices are depicted in Fig. 1d and Table 1. As was expected to produce the low-lying HOMO level of TTz containing polymers, terpolymer-based devices obtained higher $V_{oc}$ from 0.87 to 0.90 V than PM6-based device (0.86 V). Without any post-treatment, the devices exhibit the best PCE of 15.6 %, 16.5 %, 13.9 %, and 6.9 % for PM6:Y6, PM1:Y6, PM2:Y6 and PBFTz:Y6 device, respectively. However, among them, the best FF of 0.74 still has a lot of room for improvement, which limits the device performance. Fortunately, after treating with 0.75% CN, both the $J_{sc}$ and FF are remarkably improved, especially for PM1-based device, an outstanding FF of 0.78 was achieved, while the FFs of the PM6, PM2, and PBFTz-based devices are 0.72, 0.69, and 0.59, respectively. Benefiting from the significantly increased FF, the PM1-based device obtained the champion PCE of 17.6% with a small $E_{loss}$ of 0.46 eV, which is among the best values for OSCs. However, further increasing the content of TTz in the terpolymers simultaneously reduced $J_{sc}$ and FF of their corresponding OSCs.

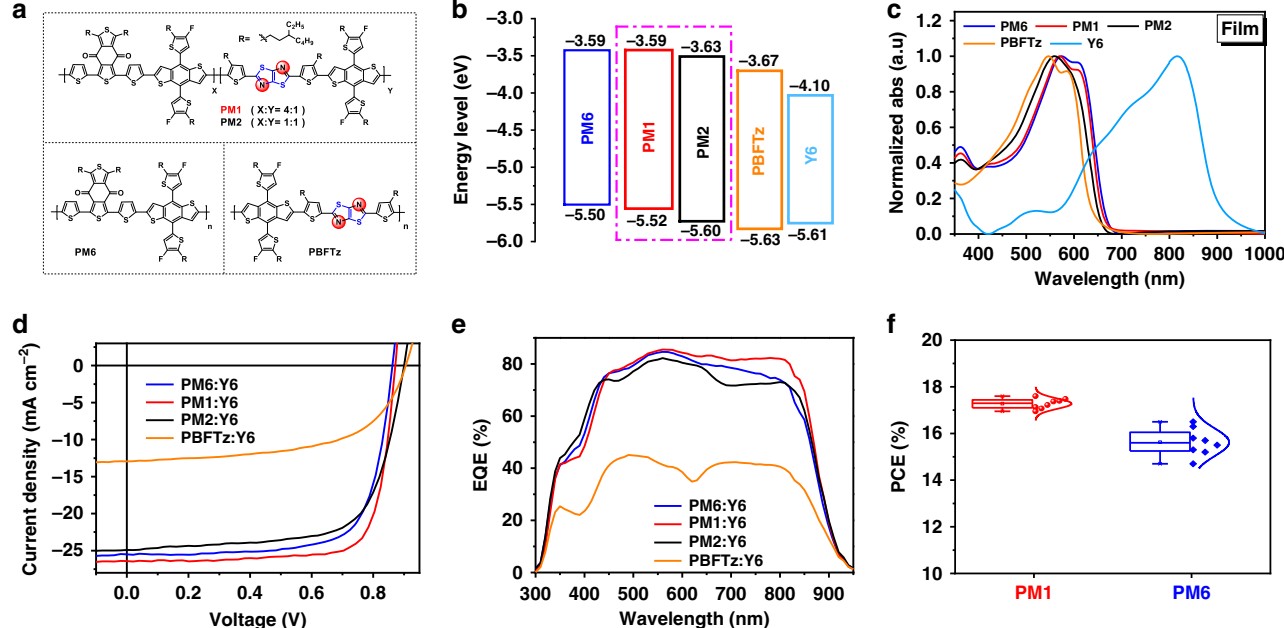

**Fig. 1 Chemical structures and optical and electrochemical and photovoltaic properties. a** Chemical structures of the donor polymers. **b** Cyclic voltammetry-derived energy level diagram. **c** UV-vis absorption spectra of donor polymers and Y6 neat films. **d** The J–V characteristics of OSCs with different polymers that have various TTz contents. **e** The EQE spectra of the OSCs. **f** Statistical histograms of PCEs measured for polymer: Y6-based cells with different polymer batches.

**Table 1 The photovoltaic parameters of OSCs with different polymers under the illumination of AM 1.5 G, 100 mW cm⁻².**

| Devices[a] | $V_{oc}$ (V) | $J_{sc}$ (mA cm⁻²) | Cal. $J_{sc}$ (mA cm⁻²)[b] | FF | PCE (%)[c] |
|---|---|---|---|---|---|
| PM6:Y6 | 0.86 | 25.5 | 25.3 | 0.72 | 15.8 (15.6 ± 0.13) |
| PM1:Y6 | 0.87 | 25.9 | 25.8 | 0.78 | 17.6 (17.3 ± 0.16) |
| PM2:Y6 | 0.90 | 24.9 | 24.1 | 0.69 | 15.5 (15.2 ± 0.17) |
| PBFTz:Y6 | 0.91 | 13.0 | 12.9 | 0.59 | 6.9 (6.7 ± 0.15) |

[a]0.75% CN.
[b]The integral $J_{sc}$ from the EQE curves.
[c]The average values and standard deviations of the device parameters based on 20 devices are shown in brackets.

The corresponding external quantum efficiency (EQE) spectra of the optimized devices with or without CN were measured to confirm the accuracy of the PCE measurements. As shown in Fig. 1e, all the EQE spectra of the devices cover a broad spectral response range from 300 to 930 nm, which is in agreement with the corresponding good complementary absorption spectra of the polymers and Y6. For the PM1-based devices, the EQE responses exceed 80% in the range of 500–810 nm, and the maximum EQE of 84% is achieved at 560 nm, which is superior to the values of the other three devices. Accordingly, the highest $J_{sc}$ of 25.8 mA cm⁻² is integrated from the EQE spectra of the PM1-based device, which also exceeds the values for the PM6 family polymers and Y6 based devices. Meanwhile, the $J_{sc}$ values calculated by integrating the EQE data agree well with those obtained from the J–V measurements within 3% mismatch.

Moreover, we also got a high certificated PCE of 17.0% from the National Institute of Metrology, China (NIM), and the optimal device can retain 90% of the original PCE for 30 days (Supplementary Figs. 9 and 10). In addition, the PCEs statistical histograms of the OSCs based on PM1 and PM6 with different batches are shown in Fig. 1f and Supplementary Table 5. The relative standard deviation, which is the ratio of standard deviation/ average PCE, was applied to estimate the reproducibility degree. The relative standard deviation value of 0.037 of the PM6-based

OSCs is three times bigger than that (0.012) of the PM1-based OSCs, indicating that PM1 exhibits the better reproducibility and the great potential to mass production for commercial application. Furthermore, to confirm the application of PM1 matching with other acceptors, we selected efficient N3[9] as electron acceptor with a device architecture of glass/ITO/PEDOT:PSS/active layer/PFN-Br/Ag. The device based on PM1:N3 yields a good PCE of 17.1%, with a $V_{oc}$ of 0.88 V, a $J_{sc}$ of 25.8 mA cm⁻², and an FF of 0.75, which is relatively higher than the reported PM6:N3-based devices (Supplementary Table 6).

**The effects of the TTz unit on polymer properties and morphology.** First, we conducted DFT calculations (at B3LYP/6-31G (d, p) basis set) to get better understanding molecular geometry of the TTz unit compared to that with the BDD unit. As shown in Fig. 2a, it is clear that the TTz unit exhibits a near-perfect coplanar molecular geometry with a dihedral angle of 0.04° between the thiazolothiazole and thiophene unit. In contrast, the BDD unit exhibits a large dihedral angle of 16.0°. The near-perfect coplanar structure of the TTz unit should be beneficial in enhancing intermolecular interaction of the polymer in films and increase the charge transport and fill factors in the device.

Next, we investigate the impacts of the TTz unit contents on the miscibility property of the polymer donor with the acceptor

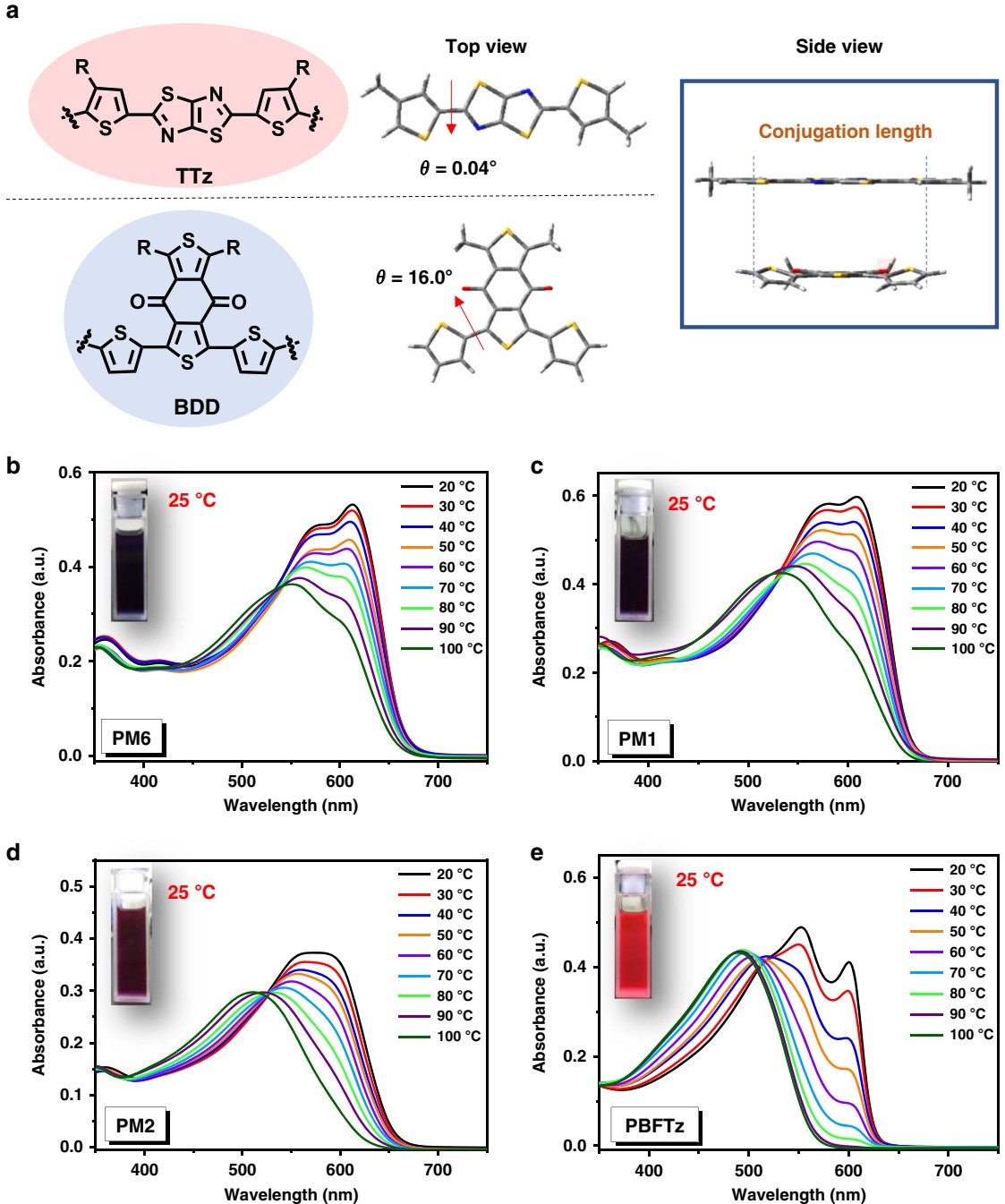

**Fig. 2 The temperature-dependent aggregation property. a** Optimized molecular conformations of TTz and BDD by DFT calculations at the B3LYP/6-31G (d, p) basis set with simplified alkyl substituents. **b**–**e** Optical absorption of the polymers in chlorobenzene at various temperatures.

by measuring the surface tension of each material via contact angle measurements[42,43]. The contact angles of two liquids (deionized water and diiodomethane) on the various neat films were measured and the results are shown in Supplementary Fig. 11. It was noted that the surface energies of the polymer donors decrease gradually from PM6 to PBFTz, which appears to be related to the increasing content of TTz in polymer skeleton. We then calculate the Flory-Huggins interaction parameter ($\chi$) for blend to show the binary miscibility from $K(\gamma_D^{-2} - \gamma_A^{-2})^2$, where $\gamma$ is the surface energy of the material, $K$ is the proportionality constant[44]. As shown in Supplementary Table 7, the $\chi$ values for Y6-based blend are increasing from PM6 to PBFTz, which will lead to a higher tendency of phase separation

and lower miscibility of mixing. Meanwhile, PM1 shows higher $(\gamma_D^{-2} - \gamma_A^{-2})^2$ value (~0.262) than that of PM6-based blend (~0.199), which should lead to a higher phase purity and contribute to the observed higher FF in the PM1-based device.

Next, we study the effect of the TTz unit on the TDA property of the polymer. As it has been reported, many high-performance donor polymers exhibit the TDA property, which is important for achieving an optimal blend morphology[45–48]. The typical TDA property of PM6 can be characterized by the temperature-dependent UV−Vis absorption spectra of the polymer solution (from 20 to 100 °C) in chlorobenzene as shown in Fig. 2b. At a high temperature of 100 °C, the absorption of the PM6 solution is peaked at 550 nm. When the solution is cooled down to room

temperature, the absorption peak red-shifts to ~570 nm and an additional 0–0 transition peak gradually raises at ~615 nm. It is important to note that the 0–0 transition peak at 615 nm is even higher than the peak at 570 nm, which indicates the strong aggregation and $\pi-\pi$ stacking of the polymer chains. Therefore, the 0–0 transition peak can be used as the "indicator" to judge whether the donor polymer has a suitable TDA property similar to that of PM6.

Figure 2c–e show the temperature-dependent UV-Vis spectra of the donor polymers. It can be seen that the TDA properties of the polymers can be changed upon the increasing of TTz content in the polymer backbone. For instance, the absorption spectrum of PM1 (20% TTz) can maintain a significant 0-0 transition peak similar to that for PM6, but the 0-0 transition park at 615 nm is not visible in the spectrum for PM2 (50% TTz). For the polymer with 100% TTz, the TDA property is completely different from PM6, which could completely change the morphology of the blends. From these results, it appears that the TTz unit does change the TDA property of the donor polymer, but using only 20% of TTz unit (in PM1), the desirable TDA property of PM6 can be mostly reserved, which is an important factor contributing to the great performance of the PM1 polymer.

Next, we investigate the effects of TTz unit on film crystallinity and charge mobility. The film crystallinity and nanostructure packing of the neat polymers and their blends were employed by the 2D grazing incident wide-angle X-ray scattering (GIWAXS) technique[49,50]. The 2D GIWAXS diffraction patterns and line cut profiles of neat films are shown in Fig. 3, with a summary of the packing parameters given in Supplementary Tables 8 and 9. Four polymers were of good crystalline order and showed a predominant "face-on" orientation relative to the substrate with obvious (010) diffraction peak at ≈1.67 Å$^{-1}$ with a $d$-spacing ~3.76 Å in the out of plane (OOP) direction and (100) diffraction peak at ≈0.30 Å$^{-1}$ with a $d$-spacing ~20.76 Å along its in plane (IP) direction. From PM6 to PBFTz, the slight reduction of $\pi-\pi$ stacking distance (3.78 Å for PM6 and PM1, 3.76 Å for PM2, and 3.74 Å for PBFTz) is observed owing to the rigid and co-plane geometry of TTz enhanced inter-chains interaction, which means that the planarity and ordering of terpolymer backbone are well maintained. Nevertheless, terpolymers still show a reduced crystallinity due to the irregular sequence of backbone. In detail, by incorporating 20% or 50% TTz unit, the crystal coherence length (CCL) values of the (100) peaks in the IP direction are obviously lower than that of PM6[51].

For morphology characterizations of the blend films, the highest intensity (010) (at ≈1.73 Å$^{-1}$) and (100) (at ≈0.30 Å$^{-1}$) diffraction peak are likely assigned to the $\pi-\pi$ stacking distance of Y6 (Supplementary Fig. 12) and lamellar stacking distance of polymer donor, respectively, where the (010) peak (at ≈1.67 Å$^{-1}$) of polymer donor and the (100) peak (at ≈0.30 Å$^{-1}$) of Y6 were hidden. With or without TTz component, the $\pi-\pi$ stacking distance in the OOP direction was barely influenced and the (100) lamellar stacking distance showed obvious decrease with decreasing steric stabilization. The degree of crystal and molecular packing in blend films is clearly changed from that of corresponding neat film discussed above. Surprisingly, in the IP direction, PM1 blend film showed larger CCL and enhanced peak intensity of (100) peak compared to PM6 by loading the optimal content of TTz unit, indicating that the crystallinity was enhanced and the density of the crystalline domains was increased. The improved crystallinity could lead to improving carrier transport, thus giving rise to elevated FF.

In order to investigate the charge transport properties of the polymers in both neat films and blends with Y6, the charge carrier mobilities were measured by space charge limited current (SCLC) method[52], and the values are listed in Supplementary

Table 10. As shown in Fig. 4a, the four polymers display a high hole mobility ($\mu_h$) in the order of $10^{-4}$ cm$^2$ V$^{-1}$ s$^{-1}$. Because of the synergy of remained crystallinity from parent polymer and enhanced conjugation in TTz, among them, PM1 shows a highest $\mu_h$ of $7.11 \times 10^{-4}$ cm$^2$ V$^{-1}$ s$^{-1}$, thus promising the increased $J_{sc}$ and FF values of the PM1-based OSCs. The mobility values show the same trend in the blend films. As expected, the PM1:Y6 blend enabled by the improved electron transport showed the highest and more balanced hole and electron transport ($\mu_h/\mu_e = 1.18$), which should originate from the optimized morphology with the increased crystallinity as shown by GIWAXS. The enhanced and well-balanced transport could effectively suppress the accumulation and recombination of charge, resulting in the improved FF of devices.

To briefly sum up the effects of the TTz unit on polymer properties, the addition of the TTz unit can increase the chi value (meaning reduced miscibility and more pure domains) and slightly increase the crystallinity and charge mobility of the donor polymer (due to the co-planar molecular geometry of TTz). Despite of the beneficial effects of the TTz unit, it can also negatively impact on the TDA property of the PM6 donor polymer. By monitoring the temperature-dependent UV-Vis absorption spectra of the polymer solutions, it is clearly seen that the polymer with 20% TTz content can maintain the optimal TDA property of the PM6 while introducing the additional benefits of TTz.

**Exciton dissociation and charge recombination**. In order to further investigate the devices on the basic operational mechanism, photocurrent density ($J_{ph}$) is plotted as a function of the effective voltage ($V_{eff}$) for four optimal OSCs[53]. As shown in Fig. 4b, under the short-circuit and maximum power output conditions, the exciton dissociation probabilities $P(E,T)$ were determined with the values being 97.2%/84.5%, 98.2%/84.5%, 96.6%/78.1%, and 92.7%/67.4% for PM6, PM1, PM2, and PBFTz, respectively. The PM1:Y6-based device shows lowest charge recombination and largest $P(E,T)$ relative to the other three devices, indicating that the PM1-based device has more efficient exciton dissociation and charge collection efficiencies, which is the main responsibility for its high $J_{sc}$ and FF as well as the best device performance.

To get insights into the effect of TTz content on the charge recombination process in the devices, the correlation between $J_{sc}$, $V_{oc}$, and light intensity ($P_{light}$) were studied (Fig. 4c, d). The $S$ values in the power-law equation $J_{sc} \propto P_{light}{}^S$ were determined to be 0.961, 0.973, 0.959, and 0.899 for PM6, PM1, PM2, and PBFTz based devices, respectively. A linear dependence of log $J_{sc}$ on log $P_{light}$ with $S$ value close to 1 implies weak bimolecular recombination in the device[54]. The $S$ value of the PM1:Y6 device comes closer to unity than others, indicating the substantially suppressed bimolecular recombination and contributing to its highest FF value. Additionally, the slope of $V_{oc}$ versus ln ($P_{light}$) for the PM6:Y6, PM1:Y6, PM2:Y6, and PBFTz:Y6 cells are 1.15, 1.07, 1.33, and 1.57 kT/q (where $k$ is the Boltzmann's constant, $T$ is the temperature in Kelvins, and $q$ is the elemental charge), respectively. In principle, the slope closes to 1.0 kT/q if the trap-assisted recombination is suppressed[55]. Hence, PM6:Y6 and PM1:Y6 devices have limited trap-assisted recombination, especially for PM1-based devices. These results are consistent with the higher $J_{sc}$ value in PM1:Y6-based device.

**Morphology characterization**. In order to understand the high FF appeared in the terpolymer-based device, we investigated the effects of TTz-based random terpolymerization strategy on surface and bulk morphology of the photoactive layers via atomic

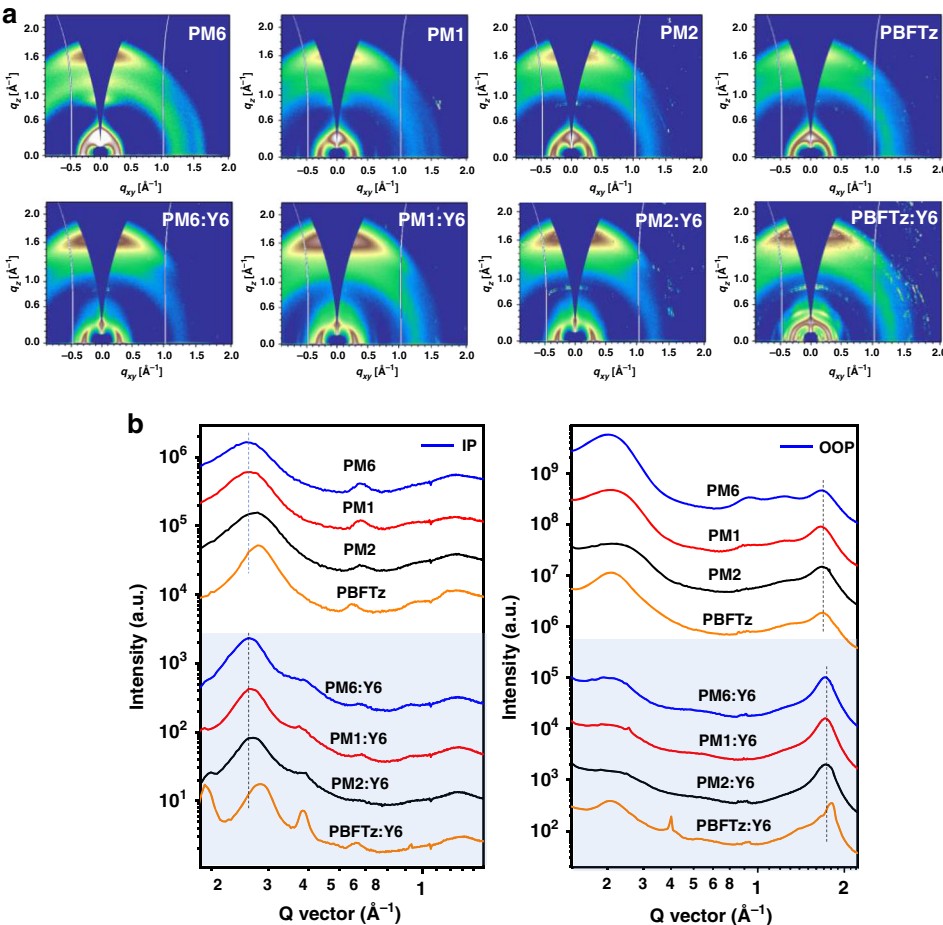

**Fig. 3 Molecular packing properties. a** 2D GIWAXS patterns for neat polymers and blend films. **b** Scattering profiles for neat polymers and blend films under optimal conditions.

force microscopy (AFM) and transmission electron microscopy (TEM), respectively. As shown in Supplementary Fig. 13, the PM6 and PM1 blend films display obviously thin fibrous structures with a relatively smooth surface with small root-mean-square (RMS) values of 0.84 and 0.93 nm. By comparison, the RMS values of PM2 and PBFTz blend films are 1.23 and 2.29 nm, respectively, which could be caused by the reduced miscibility between the polymer donor and Y6. The RMS roughness values observed in AFM images support the phase separation of all blend films in corresponding TEM images (Supplementary Fig. 14). PM6 and PM1 blend films show well-distributed fibrous structures, while the terpolymers with more TTz contents exhibit wide nano-rods and an undesirable domain size in active layers. Among them, PM1-based blend film formed more obvious phase separation in bicontinuous interpenetrating network than that of PM6, which is favorable for charge transport, as proven by the high $J_{sc}$ and FF.

## Discussion

In summary, we have successfully demonstrated over 17% efficiency in polymer solar cells by using random ternary polymerized strategy. Upon systematic variation of the thiophene-thiazolothiazole (TTz) ratio, modulation of the energy level resulted in deep HOMO energy level was observed downing from PM6 (0% of TTz) to PBFTz (100% of TTz), which enabling a higher $V_{oc}$ in the device. Owing to the enhanced rigidity and conjugation of TTz, the resulted terpolymer PM1 well restricts the adverse effect from

irregular backbone on molecular stacking and charge transport, thus conserving the high crystallinity and mobility. Besides, it also obtained relatively poor miscibility with acceptor via surface energy tuning. Therefore, PM1:Y6 blend enabled a favorable morphology of reasonably polymer domains without sacrificing polymer crystallinity in the active layer. The desired nanoscale blend morphology with favorable phase separation and good crystallinity, benefiting the exciton dissociation, charge transport, and reduced bimolecular recombination, greatly contribute to the high $J_{sc}$ and a high FF of 0.78. As a result, a high PCE of 17.6% was achieved for single-junction OSCs. It is also worth mentioning that PM1 polymer exhibits more excellent batch-to-batch reproducibility. This study demonstrates that TTz is a very effective building block for tuning the optoelectronic property and easy optimization of morphology, and PM1 polymer has a promise to become the "workhorse" material for the field of OSCs and potentially other organic electronics too.

## Methods

**Materials**. The monomers of BDT, BDD, and Y6 were purchased from Solarmer Materials Inc. TTz was synthesized in house according to the recipes reported in previous papers[35]. The preparation of the polymers was conducted using conventional Stille coupling reaction, and the detailed procedures are described in the Supplementary Information. $^1$H NMR spectra of polymers were measured in $CD_2Cl_2$ on Bruker AV 400 MHz FT-NMR spectrometer at 80 °C. Elemental analysis was conducted on a flash EA1112 analyzer.

**Device fabrication and measurement**. All the polymer-based OSCs were fabricated with a conventional device structure of glass/indium tin oxide (ITO) /poly

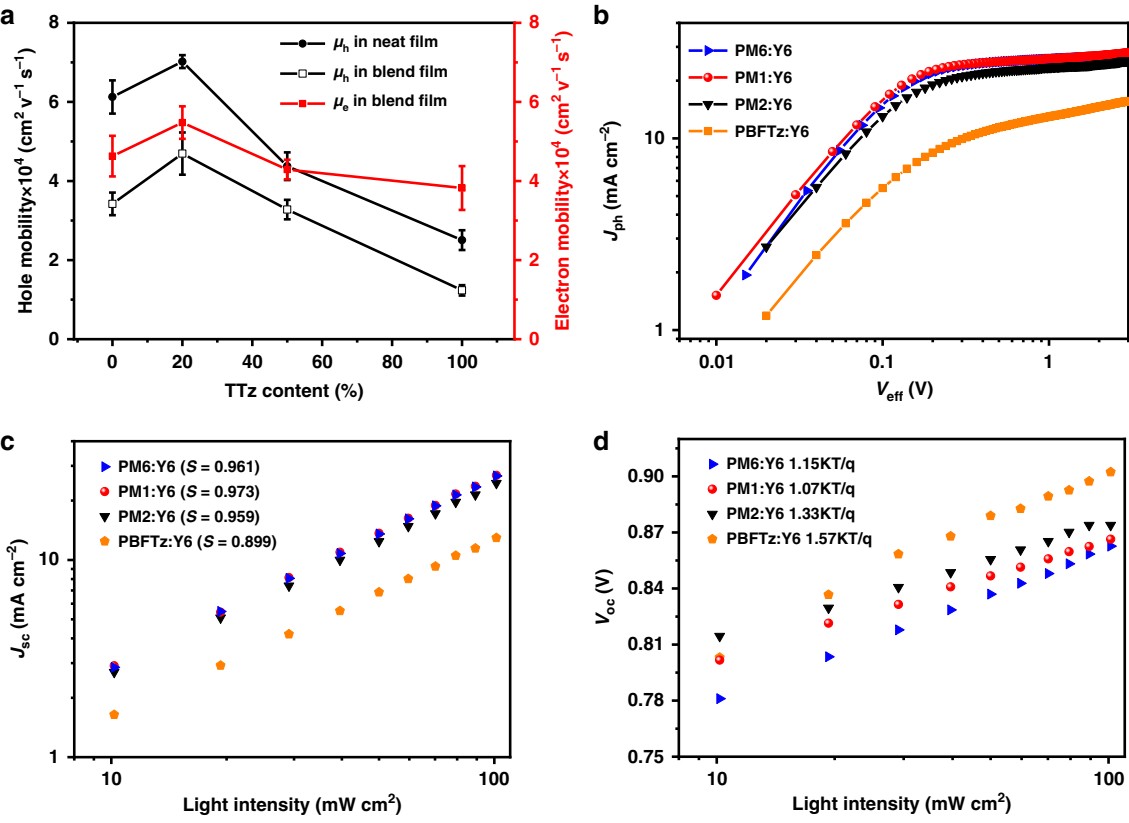

**Fig. 4 The electronic characterizations. a** Electron and hole mobilities of neat and blend films. **b** $J_{ph}$ versus $V_{eff}$ curves. **c** Light intensity dependence of $J_{sc}$. **d** Light intensity dependence of $V_{oc}$.

(3,4-ethylenedi oxythiophene): poly(styrenesulfonate) (PEDOT:PSS) /polymer:Y6/ poly[(9,9-bis(3′-(N,N-dimethyl)-nethylammoinium-propyl)-2,7-fluorene)-alt-2,7-(9,9dioctylfluorene)]dibromide (PFN-Br) /Ag. The ultrasonic cleaning was used for the ITO-coated glass with deionized water, acetone, and isopropanol, followed by drying at 100 °C for 15 min and then treated by UV-ozone for 20 min. The PEDOT:PSS (Bayer Baytron 4083) layer was deposited by spin-coating at 6000 rpm for 40 s on the ITO substrate and then thermal annealing treatment under 150 °C for 15 min. In a nitrogen glovebox, the active layers (~100 nm) based on Polymer: Y6 (1:1.25, w/w) were then deposited at top of the PEDOT:PSS layer by spin-coating a chloroform solution (total concentration of 20 mg mL⁻¹, dissolved for 5 h under 40 °C) with addition of 0.75% volume ratio of CN. The optimal speed based on PM6 (PM1, PM2):Y6 is 3500 rpm for 35 s. The optimal speed based on PBFTz: Y6 is 2000 rpm for 35 s. Then, the ultra-thin PFN-Br (0.5 mg mL⁻¹) layer was deposited by spin casting methanol solution (at 2500 rpm for 30 s), Finally, a thickness of ca. 200 nm silver was evaporated on the PFN-Br layer in vacuum at a pressure of ca. $4 \times 10^{-4}$ Pa, and the active area of the devices was 0.04 cm² determined by a shadow mask. Under the illumination of AM 1.5G (100 mW cm⁻²), the efficiencies of the OSCs were evaluated by using a SSF5-3A solar simulator (AAA grade, 50 × 50 mm² photobeam size). The SSF5-3A solar simulator and monocrystalline silicon reference cell (2 × 2 cm²) with KG5 filter (SRC-00019) were both from Enli Technology CO. Ltd. The external quantum efficiency was measured by Solar Cell Spectral Response Measurement System QE-R3011 of Enli Technology CO. Ltd. A standard single-crystal Si photovoltaic cell was used to calibrate the light intensity at each wavelength.

**UV-Vis absorption and cyclic voltammetry.** Absorption spectra of materials were performed on a UV-Vis-NIR Spectrophotometer (Agilent Technologies Carry 5000 Series). CV was recorded on a Zahner Zennium IM6 electrochemical workstation with a three-electrode configuration in Bu₄NPF₆ acetonitrile solutions (0.1 mol L⁻¹) at a scan rate of 50 mV s⁻¹.

**Contact angle measurement.** Contact angles of distinct solvents (deionized water and diiodomethane) on polymer donor and Y6 films were measured by using Dataphysics-OCA20 Micro surface contact angle analyzer. The surface tension of polymer donor and Y6 were calculated using the Owens and Wendt equation: $(1 + \cos\theta)\gamma_{pl} = 2(\gamma_s^d \gamma_{pl}^d)^{1/2} + 2(\gamma_s^p \gamma_{pl}^p)^{1/2}$, where $\gamma_s$, $\gamma_{pl}$, superscripts $d$ and $p$ are the surface energy of the sample, the probe liquid, and the dispersion and polar components of the surface energy, respectively[56].

**GIWAXS characterization.** 2D GIWAXS measurements with 10 keV X-ray beam were conducted at beamline 7.3.3 at the Advanced Light Source. Samples were prepared by spin-coating identical chloroform blend solutions as those used in OSCs on Si substrates. The grazing angle were selected at 0.12°–0.16°, maximized the scattering intensity from the samples. The CCL was calculated by CCL = 0.9 × (2π/FWHM) (Å)[47], where FWHM refers to the full width at half maximum of the corresponding diffraction peak.

**Mobility measurement.** The hole and electron mobilites were measured by using the space-charge-limited current (SCLC) method. The structures of hole-only devices and electron-only devices are ITO/PEDOT: PSS/polymer donor or active-layer/ MoO₃/Ag and ITO/ZnO/active-layer/PFN/Al, respectively. The SCLC is described by:

$$J = 9\varepsilon_0 \varepsilon_r \mu (V_{appl} - V_{bi} - V_s)^2 / 8L^3$$

where $J$ is the current density, $\varepsilon_0$ is the permittivity of free space of $8.85 \times 10^{-12}$ F m⁻¹, $\varepsilon_r$ is the relative permittivity of the material (assumed to be 3), $\mu$ is the carrier mobility, $V_{appl}$ is the applied voltage, $V_{bi}$ is the built-in voltage, $V_s$ is the voltage drop from the substrate's series resistance ($V_s = IR$), and $L$ is the active-layer thickness. The SCLC devices were measured under dark condition in a nitrogen glovebox without encapsulation.

**AFM and TEM characterization.** AFM was performed by Dimension 3100 (Veeco) Atomic Force Microscope at tapping mode. TEM was measured by Tecnai G2 F20 S-TWIN instrument (accelerating voltage, 200 kV), where the polymer:Y6 films were prepared as follow: the polymer:Y6 films were spin coated on the PEDOT:PSS-based substrates and then were immersed in deionized water to obtain floated BHJ films, and unsupported 200 mesh copper grids was used to pick films up.

**Reporting summary.** Further information on research design is available in the Nature Research Reporting Summary linked to this article.

## Data availability

All relevant data supporting the findings of this study are available from the authors on reasonable request. In addition, the source data underlying Figs. 1c–f, 2b–e, 3b, c, and 4, and Supplementary Figs. 2, 3, 4, 6, 7, 8, 10, and 12b, as well as Table 1 and Supplementary Tables 2, 3, 4, 5, and 6 are provided as a Source Date file. Source data are provided with this paper.

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

## Acknowledgements

This work was supported by National Natural Science Foundation of China (NSFC) (No. 51773142 and 51973146), the Jiangsu Provincial Natural Science Foundation (Grant No. BK20190099), Collaborative Innovation Center of Suzhou Nano Science & Technology, the Priority Academic Program Development of Jiangsu Higher Education Institutions.

## Author contributions

M.Z. conceived and developed the idea. M.Z. and H.Y. supervised the project. J.W. designed the experiments and performed chemical synthesis and properties characterization of all polymers; G.L. and X.G. carried out the device fabrication and characterization; J.F. performed efficiency certification and TEM measurements; L.Z. and F.L. performed GIWAXS measurements and analyzed the data. Y.W. provided suggestion on synthesis. M.Z., J.W., and H.Y. prepared the paper. B.G., G.Z., L.A., and Y.L. contributed to revision of the manuscript. All authors commented on the paper.

## Competing interests

The authors declare no competing interests.
