## [Peer Review File · Nature Communications]

Reviewers' comments:

Reviewer #1 (Remarks to the Author):

In this manuscript, the authors designed a series of random terpolymers for OSC applications. Among them, by introducing 20% of TTz unit into the PM6 polymer backbone, the resulting polymer (PM1) can achieve a high PCE of 17.6%. This is certainly a breakthrough in the field. In order to provide more insights into the design of high-performance polymer donor, I would suggest the authors to consider the following questions in their revised version:

1. The authors claimed that the random polymer such as PM1 can exhibit excellent batch-to-batch reproducibility. I am wondering how the authors confirm the amount of TTz content in corresponding terpolymers.
2. The authors claimed that "the downshifted HOMO energy level is attributed to the electronegativity of TTz unit". Do the authors have solid data to prove it? In addition, "TTz unit can also exhibit a near-perfect co-planar molecular geometry by DFT calculations, which is benefit for charge mobilities". This result does not seem to be consistent with the data of hole mobility and GIWAXS. How can the authors rationalize these statements? Does the 2-ethyl hexyl side chain in thiophene of backbone have any effect on the carrier transport or aggregation?
3. The efficiency is certainly very high in single junction solar cells. With the improvement of the efficiency, the OPV has now reached a stage where the stability is also calling for attention. Although the stability is not the focus of this ms, I would encourage the authors to show the thermal stability/shelf life of the optimized device?
4. From 2 D GIWAXS data, the CL of PM6 is larger than PM1. But from Figure S8, a much higher crystallinity is observed in PM1:Y6 blend system than PM6:Y6. What happened when they are blended? Does it mean that the crystallinity could be increased upon the mixture of donor and acceptor materials? If so, why not in PM2:Y6 system.
5. Why is the d-spacing of PM6:Y6 film significantly different from others?
6. The authors noted that "The PM2:Y6-based device shows lowest charge recombination and largest P(E,T) relative to the other three devices", which is not consistent with the experiment. Please check if there is any mistake in this statement.

Reviewer #2 (Remarks to the Author):

This manuscript describes the synthesis, characterization and OSC performance of a random terpolymer based on thiophene-thiazolothiazole unit. The novelty of this manuscript lies in the synthesis of a random terpolymer with a very good batch-to-batch variation. The OSC performances obtained are very high, surpassing by a small margin the well-known PM6.

That being said, I do not feel that the novelty of this manuscript is high enough to warrant publication in Nature Comm. so I recommend rejection. The manuscript is scientifically sound, but it is somewhat incremental. No new chemistry or concept is presented and the world record for a solar cells efficiency is not enough to allow publication in a high-impact journal such as Nature Comm. Of course, this manuscript will be cited. I suggest the authors to submit it in a more specialized journal.

Reviewer #3 (Remarks to the Author):

n the submitted manuscript a significantly improved efficiency for a non-fullerene organic solar cell based on a random terpolymer is described. The data are comprehensive and fully consistent with the results. Although this random terpolymerization strategy of this paper is not conceptually new, the synthesized polymer PM1 exhibits greater batch-to-batch reproducibility and higher device performance (17.6%) compared to the control polymer PM6. The results are very encouraging and warrant publication in Nature Communication. I suggest publication after major revision.

Major issues to attend:

1. This paper focuses upon a comparative study between the use of PM6 and random terpolymers PM1 and PM2 using the same polymer backbone motif. However, for this comparative study to be complete, it would be nice to know how 1-chloronaphthalene also affected the morphology of the donor polymer PM6 with the small molecule acceptor Y6. Was this looked at, both in device performance and under x-ray diffraction? This needs to be addressed to compliment the high achievements of this work to be more systematic.
2. I am confused by the authors' discussion of the photovoltaic performance (Table 1), which show a FF of 72% in the PM6-based device. However, the authors discussed and analyzed a FF of 73% in the PM6-based device with 0.75% CN (Page 7, line 143). In addition, the processing parameters in the Method section should provide more details for the device fabrication, especially for the film formation of different photovoltaic systems. As far as we known, most of optimal blends with different CN additives should be also made by thermal annealing treatments. These places additional uncertainty on the morphological characterization extracted from the morphological measurements (see comment 1).
3. The J-V curves of the corresponding devices based on different processing parameters should be added in the Supporting Information.
4. The statistical report on the photovoltaic performance of the devices should be included in the manuscript. In Table 1, the processing conditions should be added.
5. The author provided the PCEs statistical histograms of the OSCs based on PM1 and PM6 with different batches. However, we didn't find the detailed molecular weight data. Actually the authors should provide a correlation between molecular weights and device performance, not batches and device performance.
6. Regarding materials characterization, the thermal properties (e.g. DSC) of the polymers should be provided to figure out the crystallization and aggregation behaviors. Apart from these, the NMR data and molecular weight images of the investigated polymers should be added in the Supporting Information.
7. The absorption coefficients of neat polymer films should be provided. In addition, I am not sure that the captions are identical to the corresponding curves. Why the introduction of a certain amount of TTz content leads to enhanced absorption coefficient in blend films?
8. Are there statistics for the SCLC mobility values or are they just the maximum values? I believe average values would be a better representation, so please add them in Table S9.
9. I cannot agree with the explanation on page 17 that the results of the light intensity dependent Voc characteristics indicate that the relevant devices have smaller geminate recombination loss. The results reflected the degree of trap-assisted recombination in devices.
10. The authors show the 2D GIWAXS and discuss the polymer order by d-spacings. However, it is unclear that the d-spacing corresponds to which polymer. The authors should discuss this carefully by correlating the GIWAXS data of the neat films.
11. Page 2 line 45, more acceptors should be cited here, not only Y6.
12. The article is coherently written, however, there are quite a number of grammatical discrepancies which need to be rectified. Some of which are as follows. Page 3 line 58: the full name of 'D-A1-D-A2' should be provided. Page 3 line 58: 'near 17.5%' could be 'near 17.6%'. Page 3 lines 72-74 are unclear and need clarification. Page 4 line 82: 'have' could be 'has'. Page 18 line 330, 'backbone engineering' should be changed to 'TTz-based random terpolymerization strategy'. Page 19 line 360, It

should be 'a high PCE of...'. Line 361 the space of 'batch-to- batch' should be removed. This continues throughout the document.

Response to the comments of reviewer for NCOMMS-20-04779A

We would like to thank the reviewers for spending time on this paper and providing invaluable comments which substantially helped improving the quality of the paper. The manuscript has been carefully revised according to the comments.

Reviewer 1

General comments: In this manuscript, the authors designed a series of random terpolymers for OSC applications. Among them, by introducing 20% of TTz unit into the PM6 polymer backbone, the resulting polymer (PM1) can achieve a high PCE of 17.6%. This is certainly a breakthrough in the field. In order to provide more insights into the design of high-performance polymer donor, I would suggest the authors to consider the following questions in their revised version:

Comment 1: The authors claimed that the random polymer such as PM1 can exhibit excellent batch-to-batch reproducibility. I am wondering how the authors confirm the amount of TTz content in corresponding terpolymers.

Reply: Thanks for the reviewer's comments. The actual TTz loadings on polymers were confirmed by elemental analysis, which were found to be in good agreement with the feed ratios of TTz, as shown below:

Elemental analysis calcd (%) for PM1 ($C_{66.8}H_{72.4}F_2N_{0.4}O_{1.6}S_8$): C = 66.84%, H = 6.04%, N=0.47%, S=21.35%. Found: C= 67.01%, H= 6.31%, N=0.39%, S=21.57%;

Elemental analysis calcd (%) for PM2 ($C_{65}H_{75}F_2N_1O_1S_8$): C = 66.15%, H = 6.36%, N=1.19%, S=21.71%. Found: C= 65.95%, H= 6.29%, N=1.05%, S=21.77%;

From the experimental data of N element contents for three polymers, we calculated the relative ratios of N-element for terpolymers with reference to PBFTz (Elemental analysis calcd (%) for PBFTz ($C_{62}H_{74}F_2N_2S_8$): C = 65.26%, H = 6.49%, N=2.46%, S=22.46%. Found: C= 66.72%, H= 6.99%, N=2.13%, S=20.29%). The relative values for PM1 and PM2 are 18.3% and 49.3%, respectively, which are exactly close to the theoretical values (20% and 50%) taking in account of the measurement errors.

Comment 2: The authors claimed that “the downshifted HOMO energy level is attributed to the electronegativity of TTz unit”. Do the authors have solid data to prove it? In addition, “TTz unit can also exhibit a near-perfect co-planar molecular geometry by DFT calculations, which is benefit for charge mobilities”. This result does not seem to be consistent with the data of hole mobility and GIWAXS. How can the authors rationalize these statements? Does the 2-ethyl hexyl side chain in thiophene of backbone have any effect on the carrier transport or aggregation?

Reply: We appreciate the reviewer's valuable comments. As shown in Supplementary

Fig. 3 and Fig. 4, from the results of DFT calculation and CV measurement, when introducing the thiazolothiazole unit into the backbone, the LUMO and HOMO energy levels of the polymer all exhibited gradually down-shifting, which are generally considered closely related to electron-withdrawing capability of the acceptor unit in the polymer.

Figure S.4 | Molecular energy levels and wavefunction distributions of the frontier orbitals for polymer models calculated by DFT/B3LYP/6-311G (d, p) with methyl groups in replacing alkyl substituents to simplify the calculations.

TTz unit exhibits a near-perfect co-planar structure which could help to enhance the crystallinity of polymer and charge mobility, but random polymer with the irregular polymer backbone can cause negative effects on molecular stacking and charge transport of the corresponding polymers, [*Chemistry* **13**, 3688-3700 (2007), *J. Am. Chem. Soc.* **138**, 10782-10785 (2016), *Adv. Energy Mater.* **8**, 1701405 (2018), *Chem. Rev.* **115**, 12666-12731 (2015)] overall result, the incorporation of a certain amount of TTz unit slightly improved hole mobility. In addition, for the variation of mobility of the polymers, the molecular weight is also an important factor to be considered. With the contents of TTz unit increasing, the solubility of polymers in toluene apparently decreased, resulting from the enhanced self-aggregation in relation with the better co-planar structure in TTz, and then the relatively smaller molecular weights were achieved, which are inferior for the π - π stacking that is the one of the key factors for hole mobility of the polymer. From the GIXD measurement results, we can find that the values of CL in 010 direction gradually reduced with the content of TTz unit increasing. Hence, when introducing a small amount of TTz (20%), the

polymer, PM1, exhibited the highest hole mobility with the good balance between the planarity and molecular weight.

As mentioned above, the incorporation of a certain amount of TTz unit in our work could well balance the effects on molecular stacking and charge transport of the corresponding polymers. Furthermore, for the effect of side chain in thiophene π -bridges, if we just consider the intramolecular co-planarity, the 2-ethyl hexyl side chain in thiophene is exactly inferior for the crystallinity and molecular packing and hence the carrier transport or aggregation. However, for the carrier transport and aggregation of the polymer, another key point is the molecular weight, which is essential to enhance the molecular stacking as mentioned above. Hence, we chose the thiophene with side chains as π -bridge for TTz unit.

Comment 3: The efficiency is certainly very high in single junction solar cells. With the improvement of the efficiency, the OPV has now reached a stage where the stability is also calling for attention. Although the stability is not the focus of this ms, I would encourage the authors to show the thermal stability/shelf life of the optimized device?

Reply: thanks for the reviewer's advice, we add the storage stability of the optimized PM1:Y6 devices in Supplementary Fig. 9, and revised the sentence in page 8, line 163, highlighted as, "and the optimal device can retain 90% of the original PCE for 30 days"

Figure S.9 | Storage stability of the optimized PM1:Y6 devices.

Comment 4: From 2 D GIWAXS data, the CL of PM6 is larger than PM1. But from

Supplementary Fig. 8, a much higher crystallinity is observed in PM1:Y6 blend system than PM6:Y6. What happened when they are blended? Does it mean that the crystallinity could be increased upon the mixture of donor and acceptor materials? If so, why not in PM2:Y6 system.

Reply: The crystallinity in pure film mainly depends on the molecule orientation of the polymer. However, in the blend films, we must consider another key factor, the miscibility between the polymer donor and the acceptor Y6. As shown in supplementary Figure S10 and Table S8, the addition of the the TTz unit can reduce miscibility between the polymer and Y6 and improve the domain purity, which is favorable for the crystallinity of the polymer. So the crystallinity for PM1 is enhanced. For the PM2: Y6 system, the tendency is so consistent, because the CL values in the (100) direction increased from 48.53 Å for the pure PM2 film to 72.07 Å for the blend of PM2: Y6.

Comment 5: Why is the d-spacing of PM6:Y6 film significantly different from others?

Reply: Thanks for the reviewer's comments. We have corrected this error in **Supplementary Table 9** in supplementary information our revised manuscript, the value should be **21.67**.

Comment 6: The authors noted that “The PM2:Y6-based device shows lowest charge recombination and largest P(E,T) relative to the other three devices”, which is not consistent with the experiment. Please check if there is any mistake in this statement.

Reply: Thanks for the reviewer's comments. We have corrected this mistake in Page 16, line 323 to 325 in the revision, and use “**The PM1:Y6-based device shows lowest charge recombination and largest P(E,T) relative to the other three devices**” instead of “**The PM2:Y6-based device shows lowest charge recombination and largest P(E,T) relative to the other three devices**”

Reviewer 2

General comments: This manuscript describes the synthesis, characterization and OSC performance of a random terpolymer based on thiophene-thiazolothiazole unit. The novelty of this manuscript lies in the synthesis of a random terpolymer with a very good batch-to-batch variation. The OSC performances obtained are very high, surpassing by a small margin the well-known PM6. That being said, I do not feel that the novelty of this manuscript is high enough to warrant publication in Nature Comm. so I recommend rejection. The manuscript is scientifically sound, but it is somewhat incremental. No new chemistry or concept is presented and the world record for a solar cell efficiency is not enough to allow publication in a high-impact journal such as Nature Comm. Of course, this manuscript will be cited. I suggest the authors to submit it in a more specialized journal.

Reply: Thanks for the reviewer's comments. The novelty of our work lies in the development of a new donor polymer named PM1, which achieved unique properties and unprecedented performance.

Importantly, the field of non-fullerene OSC has been dominated by a donor material named PM6 in the last 3 - 4 years. Among over a thousand papers published papers on non-fullerene OSCs, more than half of the papers were based on the PM6 polymer. Despite of its great performance and wide use of PM6, the polymer exhibits notorious reproducibility issue, which can only be produced small batches with severe batch-to batch uncertainty. Such a highly sensitive material is not the ideal option of material for the OSC academic community and certainly not acceptable to the industry. However, the polymer we develop, PM1, can achieve excellent batch-to-batch reproducibility, as well as much higher performance. With these features, our PM1 polymer has the promise to become the new "work-horse" material for the OSC community and will likely make big impact to our academic society and potentially the industry too.

Reviewer 3

General comments: In the submitted manuscript a significantly improved efficiency for a non-fullerene organic solar cell based on a random terpolymer is described. The data are comprehensive and fully consistent with the results. Although this random terpolymerization strategy of this paper is not conceptually new, the synthesized polymer PM1 exhibits greater batch-to-batch reproducibility and higher device performance (17.6%) compared to the control polymer PM6. The results are very encouraging and warrant publication in Nature Communication. I suggest publication after major revision.

Comment 1: This paper focuses upon a comparative study between the use of PM6 and random terpolymers PM1 and PM2 using the same polymer backbone motif. However, for this comparative study to be complete, it would be nice to know how 1-chloronaphthalene also affected the morphology of the donor polymer PM6 with the small molecule acceptor Y6. Was this looked at, both in device performance and under x-ray diffraction? This needs to be addressed to compliment the high achievements of this work to be more systematic.

Reply: Thanks for the reviewer's suggestion. In this manuscript, we have compared the device based on the blend films as cast and with CN treatment on page 7, line 139 to 149, “Without any post-treatment, the devices exhibit the best PCE of 15.6%, 16.5%, 13.9% and 6.9% for PM6:Y6, PM1:Y6, PM2:Y6 and PBFTz:Y6 device, respectively. However, among them, the best FF of 0.74 still has a lot of room for improvement, which limits the device performance. Fortunately, after treating with 0.75% CN, both the J_{sc} and FF are remarkably improved, especially for PM1-based device, an outstanding FF of 0.78 was achieved, while the FFs of the PM6, PM2 and PBFTz -based devices are 0.72, 0.69 and 0.59, respectively. Benefiting from the significantly increased FF, the PM1-based device obtained the champion PCE of 17.6% with a small E_{loss} of 0.46 eV, which is among the top values for OSCs. Besides that, further increasing the content of TTz simultaneously reduced J_{sc} and FF” Then, we systematically studied the relationship between the device performance under the optimized processing contain (with 0.75% CN treatment) and the molecule crystallinity and packing by GWAXS measurements on page 12-13, line 253 to 261, ie. “With or without TTz unit compound, the π - π stacking distance in the OOP direction was barely influenced and the (100) lamellar stacking distance showed obvious decrease with decreasing steric stabilization. The degree of crystal and molecular packing in blend films is clearly changed from that of corresponding neat film discussed above. Surprisingly, in the IP direction, PM1 blend film showed larger CCL and enhanced peak intensity of (100) peak compared to PM6 by loading the optimal content TTz unit, indicating that the crystallinity was enhanced and the

density of the crystalline domains was increased. The improved crystallinity could lead to improving carrier transport, thus giving rise to elevated FF”.

Comment 2: I am confused by the authors' discussion of the photovoltaic performance (Table 1), which show a FF of 72% in the PM6-based device. However, the authors discussed and analyzed a FF of 73% in the PM6-based device with 0.75% CN (Page 7, line 143). In addition, the processing parameters in the Method section should provide more details for the device fabrication, especially for the film formation of different photovoltaic systems. As far as we know, most of optimal blends with different CN additives should be also made by thermal annealing treatments. These places additional uncertainty on the morphological characterization extracted from the morphological measurements (see comment 1).

Reply: Thanks for the reviewer's comments. We have updated the data in Page 7, line 145 in the revision, and use “a FF of 0.72” instead of “a FF of 0.73”.

In addition, the detailed processing parameters for the film formation of the different photovoltaic systems were provided in the “Method” part on page 19 in the manuscript, as shown below:

The OSCs were fabricated with an conventional structure of glass/indium tin oxide (ITO) /poly(3,4-ethylenedi oxythiophene): poly(styrenesulfonate) (PEDOT:PSS) /polymer:Y6/poly[(9,9-bis(3'-(N,N-dimethyl)-nethylammoinium-propyl)-2,7-fluorene)-alt-2,7-(9,9dioctylfluorene)]dibromide (PFN-Br) /Ag. In an ultrasonic bath, the ITO-coated glass was cleaned with deionized water, acetone and isopropanol, followed by drying at 100 °C for 15min. Subsequently, the pre-cleaned ITO-coated glass substrate was treated by UV-ozone for 20 min. The PEDOT:PSS (Bayer Baytron 4083) layer was deposited by spin-coating at 6000 rpm for 40 s on the ITO substrate and then thermal annealing treatment under 150 °C for 15 min. In a nitrogen glove box, the active layers (~100 nm) based on Polymer:Y6 (1:1.25, w/w) were then deposited atop the PEDOT:PSS layer by spin-coating a chloroform solution (total concentration of 20 mg mL⁻¹, dissolved 5 h under 40 °C) with addition of 0.75% volume ratio of CN. The optimal speed based on PM6 (PM1, PM2):Y6 is 3500 rpm for 35 s. The optimal speed based on PBFTz:Y6 is 2000 rpm for 35 s. Then, the ultra-thin PFN-Br (0.5 mg mL⁻¹) layer was deposited by spin casting methanol solution (from 2000 rpm for 30 s), Finally, 200 nm Ag was evaporated on the active layer under vacuum at a pressure of ca. 4 × 10⁻⁴ Pa. and through a shadow mask to determine the active area of the devices (0.04 cm²).

Furthermore, we found the optimized condition is that without thermal annealing. TA treatment was performed as shown in Table R1, the device exhibited inferior performance related to the device without TA, the V_{oc} and FF of the devices are decreased with slowly improved J_{sc}.

Table R1. Optimization of thermal annealing (TA) for PM1-based device.

Treatment details		V_{oc} (V)	J_{sc} (mA/cm ²)	FF (%)	PCE (%)
1:1.25 (w/w) 0.75% CN	Without TA	0.87	25.9	78	17.6
	80□+10min	0.86	26.3	76	17.2
	100□+10min	0.85	26.7	75	17.0
	110□+10min	0.85	27.0	73	16.8
	120□+10min	0.85	27.3	72	16.7

Comment 3: The J - V curves of the corresponding devices based on different processing parameters should be added in the supplementary information.

Reply: Thanks for the reviewer's comments. We have provided the J - V curves of the PSCs based on PM1:Y6 at different additive, different D/A weight ratios with additive (0.75 vol %) and the PSCs based on as-cast device at different polymers in Supplementary Fig. 6a-c in the revision.

Comment 4: The statistical report on the photovoltaic performance of the devices should be included in the manuscript. In Table 1, the processing conditions should be added.

Reply: Thanks for the reviewer's comments. We have added statistical diagram of PCEs for 20 polymer:Y6-based cells in Supplementary Fig. 7 and the processing conditions in Table 1 in the revised manuscript.

Figure S.7 | Statistical diagram of PCEs for 20 polymer:Y6-based cells.

Table 1 The photovoltaic parameters of OSCs with different polymers under AM 1.5 G illumination (100 mW cm^{-2}).					
Devices^{a)}	V_{oc} (V)	J_{sc} (mA cm^{-2})	Cal. J_{sc} ^{b)} (mA cm^{-2})	FF	PCE^{c)} (%)
PM6:Y6	0.86	25.5	25.3	0.72	15.8 (15.6±0.13)
PM1:Y6	0.87	25.9	25.8	0.78	17.6 (17.3±0.16)
PM2:Y6	0.90	24.9	24.1	0.69	15.5 (15.2±0.17)
PBFTz:Y6	0.91	13.0	12.9	0.59	6.90 (6.7±0.15)

a) 0.75% CN b) The integral J_{sc} from the EQE curves. c) The average values and standard deviations of the device parameters based on 20 devices are shown in brackets.

Comment 5: The author provided the PCEs statistical histograms of the OSCs based on PM1 and PM6 with different batches. However, we didn't find the detailed molecular weight data. Actually, the authors should provide a correlation between molecular weights and device performance, not batches and device performance.

Reply: We appreciate the reviewer's valuable comments. We prepared eight batches of polymer donor PM1/PM6 by controlling the polymerization reaction time about **6 h**.

The molecular weight of PM1/PM6 and corresponding photovoltaic performance of the devices are summarized in Supplementary Tables 5 and 6, and the detail synthesis of polymerization are given in the “Supplementary Note” section in supplementary information, as shown follow. With similar polymerization reaction time region, PM1 batches show M_n s from 21.2 to 33.3 kDa, while PM6 exhibit bigger different M_n s from 12.2 to 63.1 kDa. On the other hand, it is found that the medium molecular weight of the polymers leads to better photovoltaic performance. Therefore, PM1 exhibits the better reproducibility and the great potential to mass production for commercial application.

Supplementary Note:

Synthesis of PM6: BDT-TF monomer (0.3 mmol, 282 mg) and bromide monomer of BDD (0.3 mmol, 230 mg) were dissolve in toluene (10 mL). Pd(PPh₃)₄ (15 mg) was added into the mixtures after being flushed with argon for ten minutes. Then, the reaction mixtures were purged with argon for another 15 min. The reactions of eight batches of PM6 were stirred at 110°C for about 6 h. The polymers were precipitated in methanol (100 mL) and filtrated. The dried precipitates were purified by flash silica gel column chromatography by using chloroform as eluent. The polymer was then precipitated in methanol (60 mL) and dried under vacuum for 24 h before use.

Synthesis of PM1: BDT-TF monomer (0.3 mmol, 282 mg) and bromide monomers of BDD (0.24 mmol, 184 mg) and TTz (0.06 mmol, 41 mg) were dissolve in toluene (10 mL). Pd(PPh₃)₄ (15 mg) was added into the mixtures after being flushed with argon for ten minutes. Then, the reaction mixtures were purged with argon for another 15 min. The reactions of eight batches of PM1 were stirred at 110°C for about 6 h. The polymers were precipitated in methanol (100 mL) and filtrated. The dried precipitates were purified by flash silica gel column chromatography by using chloroform as eluent. The polymer was then precipitated in methanol (60 mL) and dried under vacuum for 24 h before use.

Table S5. | Summary of the molecular weight of PM1/PM6 and photovoltaic performance of PM1/PM6-based single junction cells with different batches.

Polymers	PM1			PM6		
Batches	M_n [kDa]	PDI	PCE _{max} ^{a)} [%]	M_n [kDa]	PDI	PCE _{max} ^{a)} [%]
Batch1	21.2	2.2	17.1 [16.8]	12.2	3.5	15.2 [14.9]
Batch2	23.5	2.2	17.1 [16.9]	14.8	3.6	15.3 [15.1]
Batch3	24.1	2.2	17.3 [17.0]	21.4	3.0	16.5 [16.3]
Batch4	25.6	2.3	17.2 [17.0]	21.5	2.5	16.3 [16.2]
Batch5	26.7	2.1	17.5 [17.3]	26.2	2.2	15.7 [15.6]
Batch6	28.7	2.0	17.6 [17.3]	29.7	2.5	15.8 [15.6]
Batch7	31.5	2.3	17.0 [16.8]	38.9	2.1	15.5 [15.2]
Batch8	33.0	2.3	17.4 [17.0]	63.1	2.1	14.7 [14.4]

a) The PSCs based on polymer:Y6 (1:1.25) with 0.75% CN; the statistical values in square bracket are the average PCE obtained from 10 devices.

Comment 6: Regarding materials characterization, the thermal properties (e.g. DSC) of the polymers should be provided to figure out the crystallization and aggregation behaviors. Apart from these, the ¹H NMR data and molecular weight images of the investigated polymers should be added in the Supporting Information.

Reply: Thanks for the reviewer's comments. We have investigated the crystallization and aggregation behaviors of polymers via the differential scanning calorimetry (DSC) measurement, as shown in **Supplementary Fig. 2**. We have added the comment on page 4 to 5, line 98 to 100, highlighted as, “**And the corresponding differential scanning calorimetry (DSC) measurement are shown in Supplementary Fig. 2, but there are no clear endothermic peak and exothermic peak in DSC thermogram**”. This result is consistent with literature reported PBDT-BDD-based polymers. [(a) *Macromolecules* 2012, 45(24), 9611-9617. (b) *Adv. Mater.*, 2015, 27, 4655-4660. (c)

Figure S.2 | DSC curves of polymers at a scan rate of $10\text{ }^{\circ}\text{C min}^{-1}$ under nitrogen

Furthermore, we added the ^1H NMR spectra and molecular weight images of the corresponding polymers in Supplementary Fig. 14-20, and the ^1H NMR data was provided in the “Supplementary Note” part in supplementary information, as shown below:

PM1: ^1H NMR (400 MHz, CD_2Cl_2) δ 7.20 - 6.08 (m, 7.68H), 2.75 (d, $J = 21.1$ Hz, 2H), 2.53 - 1.80 (m, 6H), 1.14 (s, 4H), 0.78 (d, $J = 71.9$ Hz, 34H), 0.59 - 0.20 (m, 24H).

PM2: ^1H NMR (400 MHz, CD_2Cl_2) δ 7.18 - 6.25 (m, 7H), 2.99 - 1.77 (m, 8H), 1.11 (d, $J = 22.9$ Hz, 4H), 0.78 (d, $J = 72.0$ Hz, 32H), 0.35 (d, $J = 43.2$ Hz, 24H).

PBFTz: ^1H NMR (400 MHz, CD_2Cl_2) δ 7.65 (d, $J = 12.3$ Hz, 2H), 7.34 (s, 2H), 7.13 (s, 2H), 2.76 (d, $J = 5.2$ Hz, 8H), 2.13 - 2.07 (m, 1H), 1.94 (d, $J = 6.0$ Hz, 1H), 1.67 (d, $J = 14.1$ Hz, 6H), 1.18 (d, $J = 14.5$ Hz, 28H), 0.93 - 0.73 (m, 24H).

Figure S.14 | ^1H NMR spectra of BDD unit.

Figure S.15 | High-temperature ^1H NMR spectra of TTz unit.

Figure S.16 | High-temperature (80°C) ^1H NMR spectra of PM1.

Figure S.17 | High-temperature (80°C) ^1H NMR spectra of PM2.

Figure S.18 | High-temperature (80°C) ^1H NMR spectra of PBFTz.

Figure S.19 | Comparison of ^1H NMR spectra of PM1(X:Y=4:1), PM2 (X:Y=1:1) and

PBFTz.

Figure S.20 | GPC traces of polymers: (a) PM6; (b) PM1; (c) PM2; (d) PBFTz. High temperature GPC with 1,2,4-trichlorobenzene as the eluent and polystyrene as a standard at 160 °C.

Comment 7: The absorption coefficients of neat polymer films should be provided. In addition, I am not sure that the captions are identical to the corresponding curves. Why the introduction of a certain amount of TTz content leads to enhanced absorption coefficient in blend films?

Reply: Thanks for the reviewer's comments. We added the absorption coefficients of neat polymer films in Supplementary Fig. 4b. Furthermore, we have corrected the mistake that the captions are not identical to the corresponding curves in Supplementary Fig 4c in the revision.

Moreover, in comparison with PM6, the introduction of a certain amount of TTz content (20%) leads to enhanced absorption coefficient in blend films, which should originate from the enhanced crystallinity of PM1 with larger coherence length as show in Figure 3 and Table S10 in supplementary information.

Figure S.4 | (b) the absorption coefficients of neat polymer films (c) UV-vis absorption spectra of the polymer:Y6 blend films.

Comment 9: Are there statistics for the SCLC mobility values or are they just the maximum values? I believe average values would be a better representation, so please add them in Table S9.

Reply: Thanks for the reviewer's comments. We have updated the data in Fig. 4a and in Supplementary Tables 1 and 10, as shown follow:

Figure 4a. Electron and hole SCLC mobilities of neat and blend films.

Table S10 | Electron and hole SCLC mobilities of blend films with 0.75% CN treatment.

Blend ^{a)}	$\mu_h (\text{cm}^2 \cdot \text{V}^{-1} \cdot \text{S}^{-1}) \times 10^4$	$\mu_e (\text{cm}^2 \cdot \text{V}^{-1} \cdot \text{S}^{-1}) \times 10^4$	μ_e/μ_h
PM6:Y6	3.68 [3.42±0.29]	5.16 [4.63±0.51]	1.40
PM1:Y6	5.02 [4.69±0.53]	5.91 [5.48±0.41]	1.18
PM2:Y6	3.51 [3.28±0.25]	4.62 [4.29±0.25]	1.32
PBFTz:Y6	1.38 [1.24±0.13]	4.19 [3.82±0.55]	3.02

a) the hole-only devices with the structure of ITO/PEDOT:PSS/polymer donor/MoO₃/Ag and the electron-only devices with the structure of ITO/ZnO/active layer/PFN-Br/Ag according to the SCLC model. b) the statistical values in square bracket are the average mobilities obtained from 4 $J^{1/2}$ - V plots.

Comment 9: I cannot agree with the explanation on page 17 that the results of the light intensity dependent V_{oc} characteristics indicate that the relevant devices have smaller geminate recombination loss. The results reflected the degree of trap-assisted recombination in devices.

Reply: We appreciate the reviewer's valuable suggestions. We have corrected this mistake in Page 16, line 323 to 325 in the revision, and use "The results indicate that PM6:Y6 and PM1:Y6 devices have limited trap-assisted recombination" instead of "The results indicate that PM6:Y6 and PM1:Y6 devices have smaller geminate recombination loss".

Comment 10: The authors show the 2D GIWAXS and discuss the polymer order by d-spacings. However, it is unclear that the d-spacing corresponds to which polymer. The authors should discuss this carefully by correlating the GIWAXS data of the neat films.

Reply: We appreciate the reviewer's valuable comments. The manuscript in page 12, line 236 to 244, was revised, highlighted as, "Four polymers were of good crystalline order and showed a predominant "face-on" orientation relative to the substrate with obvious (010) diffraction peak at $\approx 1.67 \text{ \AA}^{-1}$ with d-spacing about 3.76 \AA in the out of

plane (OOP) direction and (100) diffraction peak at $\approx 0.30 \text{ \AA}^{-1}$ with d-spacing about 20.76 \AA along its in plane (IP) direction. From PM6 to PBFTz, the slight reduction of π - π stacking distance (3.78 \AA for PM6 and PM1, 3.76 \AA for PM2 and 3.74 \AA for PBFTz) were observed owing to the rigid and co-plane geometry of TTz enhanced inter-chains interaction, which can help to maintain the ordering and planarity of the terpolymer backbone.”

Comment 11: Page 2 line 45, more acceptors should be cited here, not only Y6.

Reply: Thanks for the reviewer’s comments. We have used “Non-fullerene organic solar cells (OSCs) have been attracting increased research attentions, with the development of high-performance non-fullerene acceptors including Y6, ITIC and their derivatives” instead of “Non-fullerene organic solar cells (OSCs) have been attracting increased research attentions, with the development of high-performance non-fullerene acceptors including Y6” in Page 2 line 44 to 46. Meanwhile, we added the citation of the references in Ref [11-14]:

- 11 Lin, Y. *et al.* An electron acceptor challenging fullerenes for efficient polymer solar cells. *Adv. Mater.* **27**, 1170-1174 (2015).
- 12 Che, X., Li, Y., Qu, Y. & Forrest, S. R. High fabrication yield organic tandem photovoltaics combining vacuum- and solution-processed subcells with 15% efficiency. *Nat. Energy* **3**, 422–427 (2018).
- 13 Zhao, W. *et al.* Molecular optimization enables over 13% efficiency in organic solar cells. *J. Am. Chem. Soc.* **139**, 7148–7151 (2017).
- 14 Yao, H. *et al.* Design, synthesis, and photovoltaic characterization of a small molecular acceptor with an ultra-narrow band gap. *Angew. Chem. Int. Ed.* **56**, 3045–3049 (2017).

Comment 12: The article is coherently written, however, there are quite a number of grammatical discrepancies which need to be rectified. Some of which are as follows. Page 3 line 58: the full name of ‘D-A1-D-A2’ should be provided. Page 3 line 58: ‘near 17.5%’ could be ‘near 17.6%’. Page 3 lines 72-74 are unclear and less

clarification Page 4 line 82: 'have' could be 'has'. Page 18 line 330, 'backbone engineering' should be changed to 'TTz-based random terpolymerization strategy'. Page 19 line 360, It should be 'a high PCE of...'. Line 361 the space of 'batch-to-batch' should be removed. This continues throughout the document.

Reply: We appreciate the reviewer's valuable comments. The statements of the whole text have been carefully checked, and the sentences easy to cause misunderstanding are corrected in the revision.

We use 'Donor-Acceptor1-Donor-Acceptor2 (D-A1-D-A2)' instead of 'D-A1-D-A2' in Page 3 line 59.

We use 'near 17.6%' instead of 'near 17.5%' in Page 3 line 72.

We revise the sentence in page 3, line 71 to 73, highlighted as, 'In this article, we report a high-performance terpolymer (named PM1) that enables high OSC efficiencies (near 17.6%) and, more importantly, great device reproducibility and batch-to-batch synthetic reproducibility.'

We use 'has' instead of 'have' in Page 4 line 81.

We use 'TTz-based random terpolymerization strategy' instead of 'backbone engineering' in Page 17 line 328 to 329.

We use 'a high PCE of 17.6%' instead of 'high PCE of 17.6%' in Page 19 line 357.

We use 'batch-to-batch' instead of 'batch-to-batch' in Line 359.

REVIEWERS' COMMENTS:

Reviewer #1 (Remarks to the Author):

The authors have carefully addressed my concerns, and hence I recommend to publish the ms in Nature Communications.

Reviewer #3 (Remarks to the Author):

The authors have addressed my suggestions as well as most of those of the other reviewers. I believe the paper can be published.

Response to the comments of reviewer

We would like to thank the reviewers for spending time on this paper and providing invaluable comments which substantially helped improving the quality of the paper.

Reviewer 1**General comments:**

The authors have carefully addressed my concerns, and hence I recommend to publish the ms in Nature Communications.

Reply: Thanks for the reviewer.

Reviewer 2**General comments:**

The authors have addressed my suggestions as well as most of those of the other reviewers. I believe the paper can be published

Reply: Thanks for the reviewer.